# Long-Term Benefit of Perlingual Polybacterial Vaccines in Patients with Systemic Autoimmune Diseases and Active Immunosuppression

**DOI:** 10.3390/biomedicines11041168

**Published:** 2023-04-13

**Authors:** Inés Pérez-Sancristóbal, Eduardo de la Fuente, María Paula Álvarez-Hernández, Kissy Guevara-Hoyer, Concepción Morado, Cristina Martínez-Prada, Dalifer Freites-Nuñez, Virginia Villaverde, Miguel Fernández-Arquero, Benjamín Fernández-Gutiérrez, Silvia Sánchez-Ramón, Gloria Candelas

**Affiliations:** 1Rheumatology Department, Hospital Clínico San Carlos, 28040 Madrid, Spain; 2Rheumatology Department, Instituto de Investigación Sanitaria del Hospital Clínico San Carlos, 28040 Madrid, Spain; 3Department of Immunology, IML and IdISSC, Health Research Institute of the Hospital Clínico San Carlos (IdISSC), 28040 Madrid, Spain; 4Rheumatology Department, Hospital de Móstoles, 28935 Madrid, Spain

**Keywords:** trained-immunity-based vaccines, recurrent infections, systemic autoimmune disease prophylaxis, cohort study

## Abstract

Introduction: We have previously shown that trained-immunity-based vaccines, namely TIbV, significantly reduce the rate of recurrent infections, both of the respiratory tract (RRTI) and urinary tract infections (RUTI) in SAD patients on disease-modifying drugs (DMARDs). Objective: We evaluated the frequency of RRTI and RUTI from 2018 to 2021 in those SAD patients that received TIbV until 2018. Secondarily, we evaluated the incidence and clinical course of COVID-19 in this cohort. Methods: A retrospective observational study was conducted in a cohort of SAD patients under active immunosuppression immunized with TIbV (MV130 for RRTI and MV140 for RUTI, respectively). Results: Forty-one SAD patients on active immunosuppression that were given TIbV up to 2018 were studied for RRTI and RUTI during the 2018–2021 period. Approximately half of the patients had no infections during 2018–2021 (51.2% no RUTI and 43.5% no RRTI at all). When we compared the 3-year period with the 1-year pre-TIbV, RRTI (1.61 ± 2.26 vs. 2.76 ± 2.57; *p* = 0.002) and RUTI (1.56 ± 2.12 vs. 2.69 ± 3.07; *p* = 0.010) episodes were still significantly lower. Six SAD patients (four RA; one SLE; one MCTD) with RNA-based vaccines were infected with SARS-CoV-2, with mild disease. Conclusions: Even though the beneficial protective effects against infections of TIbV progressively decreased, they remained low for up to 3 years, with significantly reduced infections compared to the year prior to vaccination, further supporting a long-term benefit of TIbV in this setting. Moreover, an absence of infections was observed in almost half of patients.

## 1. Introduction

Traditional (non-biological) and biologic disease-modifying anti-rheumatic drugs (DMARDs) are widely used today and represent the basis of treatment for systemic autoimmune diseases (SAD). These drugs have significantly improved the prognosis of these diseases. In exchange, comorbidities and infectious complications associated with the use of DMARDs are a major concern for rheumatologists.

Rheumatic disease itself is characterized by immune dysfunction but the increased risk of recurrent infection that are due to three main factors: underlying immune dysfunction, disease activity and immunosuppression [1,2]. The use of biologic therapies increase susceptibility to recurrent or severe infections related to these immune pathways [1,2]. Different registries have been created based on a prospective follow-up of patients receiving these treatments to ascertain their long-term safety in usual clinical practice [1]. The Spanish register (BIOBADASER) found that anti-TNF therapy is associated with higher susceptibility to severe infections in rheumatoid arthritis (RA) patients in particular during the first 6 months of treatment [3]. Regarding rituximab, Curtis et al. [4] reported a higher rate of severe infections in RA patients that were treated with rituximab without biologic therapy in the previous year, compared to patients who had been previously treated with other biologic drugs (hospitalization rate for severe infection was 10.4 (6.3–17.2) vs. 7.1 (4.6–10) per 100 patient-years). Glucocorticoids may decrease the response of innate and adaptive immunity, favoring susceptibility to bacterial infections by streptococci, Staphylococcus bacterial infections by streptococci, Staphylococcus aureus, Gram-negative bacilli, as well as fungal diseases such as Candida and Aspergillus Candida and *Aspergillus* spp.

Antibiotics are the cornerstone of the treatment of infectious diseases. An urgent search is currently underway for alternatives to antibiotics to prevent infections, due to the accelerated evolution and increase in antibiotic resistance. However, there is a growing risk of multidrug-resistant infections, which becomes even higher in those patients who suffer from repeated infections, further aggravated by dysbiosis secondary to antibiotic intake [5]. Therefore, alternative or adjuvant therapies to antibiotics are increasingly needed, among which sublingual polybacterial vaccines may represent an interesting option.

Perlingual polybacterial vaccines have already been used to prevent repeated infections in immunocompetent patients [6], and subsequently used in immunocompromised patients [7]. In this setting, perlingual polybacterial vaccines have been shown to enhance trained immunity and also adaptive responses against specific antigens contained in the bacterial combination, for which they are conceptualized among the so-called trained-immunity-based vaccines (TIbV) [8,9,10,11,12,13]. TIbV represent a promising strategy in the prevention of recurrent bacterial and viral infections, reducing the use of antibiotics and the concomitant occurrence of resistances.

In contrast to classical parenteral vaccines, TIbV exert their action at locations where pathogens enter the body and initiate infections, such as mucosal surfaces, and can block transmission of the pathogen. Therefore, these vaccines are also called mucosal anti-infective vaccines.

Several randomized double-blind clinical trials and retrospective observational studies reveal data on the composition and efficacy of these vaccines [14,15].

In particular, the study previously developed by our group in patients with SAD under immunosuppressives, in which we used the MV130 vaccines for recurrent respiratory tract infections (RRTI) and MV140 for recurrent urinary tract infections (RUTI) showed a significant reduction in the frequency of RUTI and RRTI, as well as in the hospitalization rate of patients with RRTI. Moreover, antibiotic prescriptions also decreased significantly for both infections [7,16].

However, in 2018 these sublingual polybacterial vaccines were discontinued due to regulatory issues in Spain, as they were considered to follow regulatory authorization for drugs instead of that for vaccines. We aimed to determine whether these formulations administered prior to 2018 in the above-mentioned cohort had exerted a long-term effect or demonstrated long-term benefit. For this purpose, we assessed the frequency of RRTI and RUTI during the period from 2018 to 2021. Secondarily, we aim to ascertain the potential role of TIbV in diminishing the severity of COVID-19.

## 2. Material and Methods

### 2.1. Study Subjects and Design

The present study was a single-institution retrospective observational study conducted at the Hospital Clínico San Carlos, Madrid, Spain, from May 2016 to December 2021. Forty-one patients diagnosed and being followed for systemic autoimmune diseases (SAD) under active immunosuppressive treatment in the Rheumatology Department were referred due to recurrent infections to the Clinical Immunology Department for evaluation. All of them underwent immunological work-up, and were treated for recurrent infections (urinary and/or respiratory) with mucosal vaccines between 2014 and 2016 with a statistically significant reduction for both types of infections, antibiotic prescriptions and hospitalizations during the year following the administration of the vaccine; for more information, see immune intervention. Recurrent infections were considered as three or more infectious processes at the lower urinary tract. At the respiratory level, those patients with three or more infectious episodes per year, with the presence of at least one episode of pneumonia per year, were included.

Data were collected through hospital medical records and/or through direct questions during regular clinical visits.

This work was authorized by the Scientific and Ethics Committee of the hospital (16/191-E). Written informed consent for participation was waived for this study in accordance with the national legislation and the institutional requirements, given that data were collected retrospectively from medical records as per routine clinical practice.

### 2.2. Immune Intervention (Immunotherapy)

All patients were immunized for the first time between 2014 and 2016 using mucosal vaccines. The RRTI-group received a TIbV MV130 preparation (Bactek^®^, Inmunotek, Madrid, Spain) daily for 3 months. MV130 is composed of whole complete heat-inactivated bacteria, as follows: 90% Gram-positive bacteria, including *Staphylococcus* spp., *Streptococcus pneumonia*; and 10% Gram-negative bacteria, including *Klebsiella pneumoniae*, *Moraxella catarrhalis*, *Haemophilus influenzae*, each at 300 Formazin Turbidity Units (FTU)/mL (~10^9^ bacteria/mL). Likewise, the RUTI-group received the MV140 perlingual immunotherapy daily for 3 months. The proportion in MV140 is inverted, with 25% gram positive (*Enterococcus faecalis*), and 75% gram negative (*Klebsiella pneumoniae*, *Escherichia coli* and *Proteus vulgaris*). TIbV were administered perlingually by spraying two puffs of 100 μL each under the tongue, daily for 3 months. Usually, TIbV were repeated every 12-months for a maximum of 3 years.

No patient received more doses of these or another mucosal vaccines after the indicated dates. Nor did they receive any other type of immunologically based treatment with the aim of reducing the prevalence or severity of infectious processes.

### 2.3. Clinical Outcomes

The primary goal was to analyze whether the beneficial effects of the MV130 and MV140 perlingual vaccines (duration of 3 months yearly) hold up beyond one year after their use in patients diagnosed with SAD under active immunosuppressive treatment. Secondly, we aimed to determine the incidence and severity of COVID-19 in this population of patients.

### 2.4. Statistical Analysis

Continuous variables are expressed as mean ± standard deviation (SD) or median (range, min-max), depending on normal distribution, whereas frequency (%) is used for categorical data. Normal distribution of data was analyzed by means of Shapiro–Wilk test. For objective parameters of infection control, patients served as their own control, and paired data were analyzed using the paired t-test for values before and after TIbV.

For clinical variables including infections, analysis was carried out with Wilcoxon signed-rank test or McNemar test, comparison was conducted between two sets of scores that came from the same participant. SPSS V 15 and GraphPad Prism software (GraphPad Software, La Jolla, CA, USA, version 8) were used. Differences were considered statistically significant at *p* < 0.05. No additional factors were considered in the analysis.

## 3. Results

### 3.1. Epidemiological and Clinical Features

Initially, a total of 59 SAD patients with RRTI, RUTI or both were referred to the Immunology Department evaluation. Fifty-five of them were included in the study, and 41 of them completed 1-year follow-up after perlingual vaccination with TIbV (MV130 or MV140, respectively). As shown in Table 1, the mean age of patients was 54.68 years and 92.7% were females. According to the patients’ SAD diagnosis, RA represented almost half of the patients, followed by SLE (19.5%), mixed connective tissue disease (MCTD) (7.3%), and almost 20% miscellanea (Table 1). All patients had previously received the perlingual TIbV vaccine for three consecutive years from 2014 to the end of 2017 and were therefore comparable for infectious outcome.

Concerning baseline immunological status, 19.5% of patients presented antibody deficiency and less than 10% subjects hypogammaglobulinemia (Table 1).

All patients were on DMARDs at baseline, with methotrexate being the most frequent. Biological treatments were added to conventional DMARDs, with antiTNF–α being the most frequently used (38.0%), followed by anti-CD20 (14%) and tocilizumab (6.8%). Additionally, 87.2% patients were on low-dose steroids (90% on 2.5 mg daily, 10% on 5 mg daily) (Figure 1).

### 3.2. Frequency of Infections Following Suspension of Perlingual Vaccination

The frequency of infections was recorded for each patient in the pre-vaccination year, in the post-vaccination year and in the period from 2018 to 2021 (3-year period).

Globally, a statistically significant increase in the absolute number of infections was observed from 2018 to 2021 (3-year sum) when compared to 1-year post-vaccination, for both RUTI (1.53 ± 2.17 versus 0.63 ± 1.13; *p* = 0.005) and for RRTI (1.63 ± 2.32 versus 0.67 ± 0.92; *p* = 0.003).

Notably, despite the increase in the absolute numbers of RUTI and RRTI in the 3-year period, approximately half of the patients had no infections at all. Specifically, for RUTI, 52.0% of patients remained without infection (Figure 2) and 43.4% of RRTI patients (Figure 3).

When we compared the number of infections in the 3-year period with respect to the pre-vaccine year, RRTI episodes were still significantly lower (1.61 ± 2.26 versus 2.76 ± 2.57; *p* = 0.002), as well as RUTI episodes (1.56 ± 2.12 versus 2.69 ± 3.07; *p* = 0.010).

We then divided the entire cohort into two subgroups according to clinical outcomes during the 3-year follow up (patients with and without infections) and compared them (Table 2). Almost half of the patients that presented RRTI and/or RUTI remained without infections for the long term. There were no differences in terms of age, sex of underlying SAD or treatments used with respect to infections outcomes.

### 3.3. COVID-19 Incidence and Severity

Six SAD patients (4 RA; 1 SLE; 1 MCTD) were infected with SARS-CoV-2. All patients were vaccinated with RNA-based anti-SARS-CoV-2 vaccines prior to infection. All patients presented with mild COVID-19, and none were hospitalized or needed oxygen.

## 4. Discussion

Infection is one of the main causes of morbidity and mortality in patients with systemic autoimmune diseases (SAD). Predisposition to infection in patients with SAD is associated with an abnormal response of the innate immune system associated with an abnormal response of the innate and adaptive immune system underlying the disease itself and secondary to immunomodulatory and immunosuppressive therapies.

To the best of our knowledge, this is the first study addressing the long-term benefit of TIbV. We show here that mucosal TIbV based on polybacterial combinations exerts a long-term effect for up to 3 years after administration. Although the frequency of urinary and respiratory tract infections increased during the three years after immunization, half of the patients did not have intercurrent infections. The clinical beneficial effects of these vaccines remained until up to 3 years later, with a significant decrease in infections compared to the year prior to vaccine administration suggesting that these vaccines may have a long-term benefit. Only six patients presented with COVID-19, which might in part be ascribed to the enhanced trained immunity in these patients and the potential for abrogating virus expansion from oronasal mucosa, since RNA anti-SARS-CoV-2 vaccines have not been shown to diminish infection acquisition. In this context, these TIbV may suppose an advantage to the avoidance of viral spread and contagion, as has been demonstrated in experimental models with the MV130 formula [17,18]. These vaccines may represent a safe alternative in the case of emergent epidemics.

These data confirm the beneficial effects of these vaccines shown in previous studies in terms of heterologous infection protection [14,15,16]. Our data further confirm the beneficial effects shown in our previous observational study in patients with SAD under immunosuppressants [7]. Furthermore, our results reaffirm the benefits described in relation to subsequent potential decrease in the rate of antibiotic resistance with the expected improvement in the patients’ quality of life [7].

Once the administration of the vaccines was withdrawn, despite the increase in infections when compared to the post-vaccination year, a decrease in the number of infections with respect to the year prior to vaccination and even an absence of infections in up to half of the patients.

This objective long-term clinical benefit against heterologous pathogens could be explained by “trained immunity”. The protective effects of these vaccines have been shown to be non-specific against different pathogens, and also specific to the antigens contained in the preparations. This broad-spectrum effect is due to immunomodulatory activity on the innate immune response, according to several in vivo and in vitro studies [19,20].

Inactivated combinations of G-positive and G-negative bacteria contained in MV130 and MV140 seem to work synergistically by activating cells of innate immunity through metabolic and epigenetic reprogramming of these cells in an innate memory status called “trained immunity” [21,22,23]. These mechanisms enable the testing of new approaches in immunology for the development of trained immunity-based vaccines (TIbV) [9].

As a limitation, this study did not have a control group, because it was not designed to be a clinical trial.

Our results are particularly relevant for patients with recurrent infections, such as immunosuppressed patients, especially those of the respiratory tract and genitourinary tract, for which specific vaccines are currently unavailable.

In conclusion, the TIbV show beneficial effects in the long term in SAD patients under active immunosuppression. Very few SAD patients presented with COVID-19, all with mild disease. These TIbV could help to solve a common clinical problem faced by rheumatologists treating patients with immunosuppressive therapy, which adds to previous studies on non-immunosuppressed patients [24,25]. Further prospective studies and clinical trials with a larger cohort of SAD patients are needed to validate these results as well as to confirm the clinical significance and benefits of these findings.

## Figures and Tables

**Figure 1 biomedicines-11-01168-f001:**
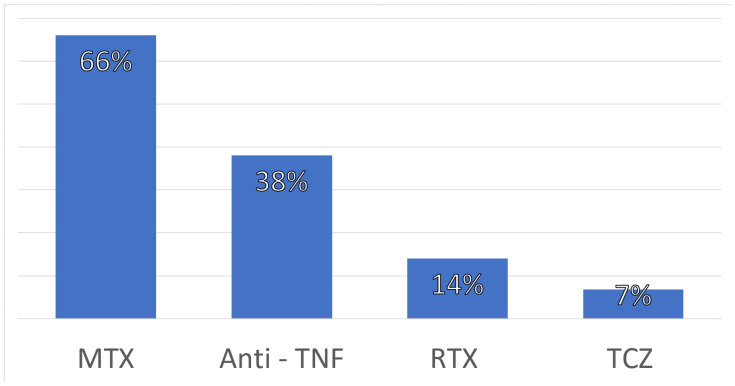
Baseline treatments in SAD patients and vaccinated with sublingual polybacterial vaccines (N 0 41). MTX: Methotrexate. Anti-TNF: anti-tumor necrosis factor. RTX: Rituximab. TCZ: Tocilizumab.

**Figure 2 biomedicines-11-01168-f002:**
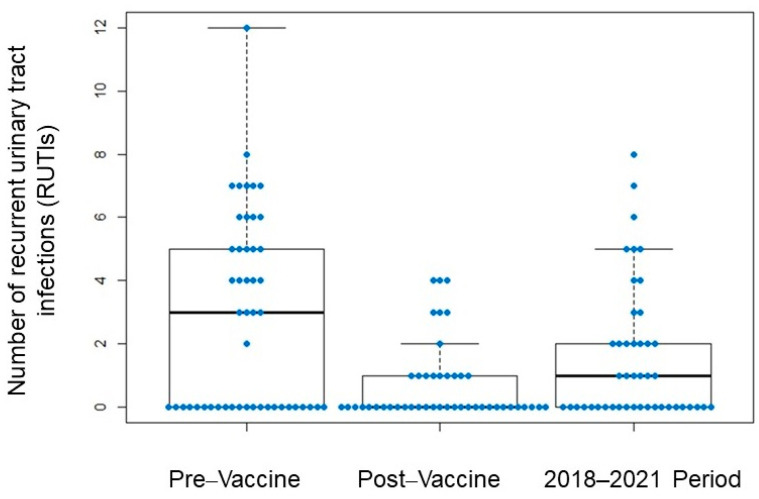
Incidence of recurrent urinary tract infections (RUTIs).

**Figure 3 biomedicines-11-01168-f003:**
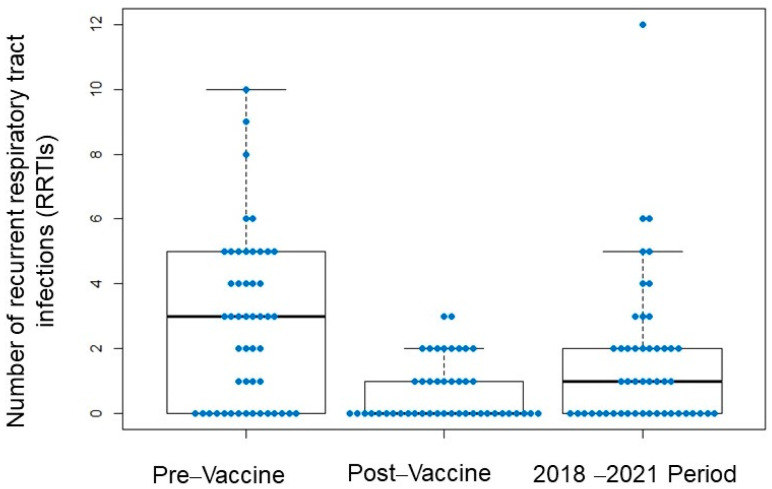
Incidence of recurrent respiratory tract infections (RRTIs).

**Table 1 biomedicines-11-01168-t001:** Demographic and clinical characteristics of subjects.

	Mean ± SD or N (%)
Demographic characteristics	
Age (years)	54.68 ± 14.8
Sex (%females)	92.7%
Rheumatic diseases	
Rheumatoid arthritis (RA)	43.9%
Systematic lupus erythematosus (SLE)	19.5%
Mixed connective tissue disease (MCTD)	7.3%
Others	29.3%
Recurrent infectious diseases	
Recurrent respiratory tract infections	34.1%
Recurrent urinary tract infections	46.3%
Both	19.5%
Inmunological status	
Antibody deficiency	19.5%
Hypogammaglobulinemia	7.3%
Others	12.2%

Data is expressed as mean ± SD or frequency (%) of total subjects.

**Table 2 biomedicines-11-01168-t002:** Comparison of the whole SAD cohort, dividing patients according to clinical outcome during the 3-year period of follow up. As shown, 14 patients overlapped RRTI and RUTI. No (%). ^1^ mean ± edm (median).

	Patients without Infections (2018 to 2022)	Patients with Infections (2018 to 2022)	*p*
	No. 17	No. 24	
RRTI (N = 30)	13 (43.4%)	17 (56.6%)	NS
RUTI (N = 25)	13 (52.0%)	12 (48%)	NS
Age (years)	55.2 ± 16.6 (54.5) ^1^	59.2 ± 12.6 (58.0) ^1^	NS
Sex (M/F)	1/17	2/24	NS
Rheumatoid arthritis	8	9	NS
SLE	8	6	NS
DMARDc *	14 (82.3%)	16 (66.6%)	NS
DMARDb **	1 (6%)	1 (4%)	NS
Ttos combinados (DMARDc + DMARDb)	2 (11.7%)	7 (29%)	NS

* DMARDc: conventional DMARD: Metotrexato; Hidroxychloroquine; Azathioprine; Mycophenolate; Leflunomide. DMARD; ** DMARDb: BiologicalDMARD: Anti-TNF; Abatacept; Tocilizumab; Rituximab. M, male; F, female. NS, non-significant.

## Data Availability

The raw data supporting the conclusions of this article will be made available by the authors, without undue reservation.

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
