# Peer review of "Long-Term Benefit of Perlingual Polybacterial Vaccines in Patients with Systemic Autoimmune Diseases and Active Immunosuppression"

_biomedicines, 2023, doi:10.3390/biomedicines11041168_

Round 1
Reviewer 1 Report
The authors report on the long-term effects of sublingual polymicrobial vaccines in systemic autoimmune diseases and active immunosuppressive states.
systemic autoimmune diseases and active immunosuppressive states.
The submission is in the category of Breif report.
Could the following points be improved?
Abstract is too long. The abstract should be cut in half.
The introducrtion of abstract and text starts with DMARDs, but this is only a group of drugs that are considered as anti-rheumatic drugs. However, the case series seems to include other autoimmune diseases in more than half of the cases. Is this not a logical development?
I don't see the point of presenting the 2018-2021period. If you want to observe long-term effects, you should consider this in terms of the number of years that have passed since the vaccine was implemented in individual cases. Also, if we are going to compare the 2018-2021 PERIOD, we would have to compare it to the 3 years before all cases were administered the vaccine.
The authors themselves should state the limitation in the discussion. The authors should also mention the future improvements for these events. For example, the number of n is 41, which is still low, and if the discussion is to start with DMARDs, the use of each of the 41 cases should be described in more detail than in Figure 1.
Author Response
Estimated Reviewer. We send you our corrections.
Sincerely
Gloria Candelas

Reviewer 2 Report
The authors submitted a research article in which they investigated the potential role of trained-immunity-based vaccines in diminishing severity of COVID-19. They included 41 patients with systemic autoimmune diseases under active immunosuppressive treatment who were immunized for the first time between with mucosal vaccines. The authors followed them for 3 years and found that trained-immunity-based vaccine show beneficial effects in the long term in SAD patients under active immunosuppression. They concluded that "These TIbV could help to solve a common clinical problem faced by rheumatologists treating patients with immunosuppressive therapy, which adds to previous studies on non-immunosuppressed patients". This fact seem to be interesting and I would like to congratulate the authors on it. However, there are some issues which deserve to be discussed.
1. The mai problematic issue is a lack of control group. The authors should devide the entire cohort on 2 subgroups according to clinical outcomes and compare them.
2. Please, add clinical characteristis of both subgroups and re-write the subsection discussion accordingly.
Author Response
Estimated reviewer. We send you our corrections.
Sincerely
Gloria Candelas

Round 2
Reviewer 1 Report
Now, authors modified theri manuscript in this revised version according to the reviewers' comments, adequately.
Reviewer 2 Report
The authors submitted a revised version of the paper alon with a clear explanation by wich the corrections were made. I have no serious concerns about the manuscript in its revised version.